

# Is coconut coir dust an efficient biofertilizer carrier for promoting coffee seedling growth and nutrient uptake?

Yupa Chromkaew[1], Thewin Kaeomuangmoon[1], Nipon Mawan[2], Nilita Mukjang[3] and Nuttapon Khongdee[2]

[1] Department of Plant and Soil Science, Faculty of Agriculture, Chiang Mai University, Chiang Mai, Thailand
[2] Department of Highland Agriculture and Natural Resources, Faculty of Agriculture, Chiang Mai University, Chiang Mai, Thailand
[3] Department of Entomology and Plant Pathology, Faculty of Agriculture, Chiang Mai University, Chiang Mai, Thailand

## ABSTRACT

**Background**. As a method for sustainable agriculture, biofertilizers containing plant growth-promoting bacteria (PGPB) have been recommended as an alternative to chemical fertilizers. However, the short shelf-life of inoculants remains a limiting factor in the development of biofertilizer technology. The present study aimed to (i) evaluate the effectiveness of four different carriers (perlite, vermiculite, diatomite and coconut coir dust) on the shelf-life of S2-4a1 and R2-3b1 isolates over 60 days after inoculation and (ii) evaluate isolated bacteria as growth-promoting agents for coffee seedlings.

**Methods**. The rhizosphere soil-isolated S2-4a1 and plant-tissue-isolated R2-3b1 were chosen based on their P and K-solubilizing capacities and their ability to produce IAA. To evaluate the alternative carriers, two selected isolates were inoculated with the four different carriers and incubated at 25 °C for 60 days. The bacterial survival, pH, and EC in each carrier were investigated. In addition, coconut coir dust inoculated with the selected isolates was applied to the soil in pots planted with coffee (*Coffea arabica*). At 90 days following application, variables such as biomass and total N, P, K, Ca, and Mg uptakes of coffee seedlings were examined.

**Results**. The results showed that after 60 days of inoculation at 25 °C, the population of S2-4a1 and R2-3b1 in coconut coir dust carriers was 1.3 and $2.15 \times 10^8$ CFU $g^{-1}$, respectively. However, there were no significant differences among carriers ($P > 0.05$). The results of the present study suggested that coconut coir dust can be used as an alternative carrier for S2-4a1 and R2-3b1 isolates. The significant differences in pH and EC were observed by different carriers ($P < 0.01$) after inoculation with both bacterial isolates. However, pH and EC declined significantly only with coconut coir dust during the incubation period. In addition, coconut coir dust-based bioformulations of both S2-4a1 and R2-3b1 enhanced plant growth and nutrient uptake (P, K, Ca, Mg), providing evidence that isolated bacteria possess additional growth-promoting properties.

Corresponding authors
Nipon Mawan, nipon-mawan93@gmail.com
Nuttapon Khongdee, nuttapon.k@cmu.ac.th

## INTRODUCTION

Coffee is one of the most significant industrial crops in Thailand, creating a revenue of more than 173.99 million USD annually. Arabica and Robusta are the most popular coffee varieties consumed worldwide. In Thailand, the bulk of Robusta coffee plantations are located in the south, accounting for 75% of the total Robusta coffee plantations, whereas Arabica coffee plants are located in the north (*Kiattisin, Nantarat & Leelapornpisid, 2016*). By focusing solely on boosting coffee production yields, the coffee policy of Thai government aims to increase the competitiveness of coffee productivity and its quality within Association of Southeast Asian Nations (ASEAN) markets. In order to improve coffee productivity and quality, fertilizer has been becoming an important factor. Although chemical fertilization has been known as a beneficial component to achieve a good crop productivity, long-term investigations have revealed its negative effects on soil fertility (*Lin et al., 2019*; *Yadav & Sarkar, 2019*). Thus, environmentally friendly agriculture is necessary for farmers with limited resources. Therefore, it is vital to discover adequate tools for sustainable coffee production in Thailand.

Several investigations have demonstrated that the coffee rhizosphere is surrounded with a wide range of microorganisms (*Caldwell et al., 2015*; *Muleta, Assefa & Granhall, 2007*; *Zhao et al., 2018*). Some microorganisms, particularly bacteria and fungi, have been shown to have the ability to improve plant performance by several mechanisms, commonly referred as "plant growth-promoting microbes" (PGPM) (*Vessey, 2003*; *Zhang & Kong, 2014*). Interactions between plants and microbes may have either a net positive or negative effect on plant growth. Recently, bacterial endophytes have received considerable attention because of their abilities to modulate plant growth and suppress plant pathogens through different mechanisms. These microorganisms exist within the living tissues of plants without causing damage to the host or evoking symptoms of plant disease (*Reinhold-Hurek & Hurek, 2011*; *Brader et al., 2014*). In contrast, these organisms indirectly promote plant growth by synthesizing phytohormones such as indoleacetic acid and gibberellins related to an increase in stem and root growth (*Lubna et al., 2018*; *Vessey, 2003*), increasing the availability of nutrients such as nitrogen (N), phosphorus (P), potassium (K) and iron (Fe) in the rhizosphere and protecting plants from soil pathogens (*Rajkumar et al., 2010*; *Vílchez et al., 2016*). Consequently, PGPM have also been proposed as critical components of new agricultural technologies and environmentally friendly agricultural systems (*Mimmo et al., 2018*; *Asghari, Khademian & Sedaghati, 2020*; *Etesami, 2020*).

Many of the PGPM have been isolated and have demonstrated their high performance under laboratory conditions. However, a lack of suitable carriers for biofertilizer, especially non-spore forming bacteria, influences the survival rates of inoculum, thereby reducing its efficiency after application into the soil. Carrier materials could provide suitable conditions for microbial colonization by building protected pore spaces and changing the soil physical characteristics (*Van Elsas et al., 1992*). Most of the biofertilizers for plant growth promoting rhizobacteria (PGPB) are formulated with solid carriers such as talc, perlite, vermiculite and peat (*Brahmaprakash & Sahu, 2012*; *Sangeetha, 2012*). Due to a scarcity of naturally occurring raw resources, some materials are unavailable in some countries. Hence, it

is necessary to develop an alternative capable carrier for retaining bacterial viability for extended periods of time without losing their effectiveness.

On this subject, coconut coir dust (CCD) has been taken into consideration as an alternative carrier utilized for the development of biofertilizers. The thick mesocarp, or husk, of the coconut fruit (*Cocos nucifera* L.), which remains after the coconut husk has been processed to produce the long fibers for industry, is the source of CCD. The mixture of CCD with other components has been widely and successfully used in many regions around the world as an environmentally friendly peat substitute for growing plants in the containers (*Meerow, 1994*). In addition, the physico-chemical characteristics of the CCD such as its enrich of nutrient content, high water holding capacity and small particle size are required for good carriers (*Smith, 1992*). Consequently, the CCD is an attractive choice as a carrier material for biofertilizers. In addition, the use of CCD as a carrier for biofertilizers could be compatible with ordinary field procedures devoted to agricultural seedlings, thereby increasing farmer adoption. However, very little is known regarding the use of CCD as a carrier for PGPB. The population of the inoculant after incorporation has been used as a primary criterion for a good carrier (*Hale, Luth & Crowley, 2015*; *Tripti et al., 2017*). Therefore, this study aimed to determine the effectiveness of coconut coir dust as a carrier to maintain the survival of two bacterial isolates compared to other commercial carriers as well as to evaluate the potential of the coconut coir dust based-bioformulations for the promotion of coffee seedling growth by determining biomass and nutrient uptake, which were determined over 90 days after inoculation.

## MATERIALS & METHODS

### Bacterial isolation from soil and root tissue of coffee rhizosphere

Rhizosphere soil and root tissue samples of coffee (*Coffea arabica* L.) were collected from abandoned coffee plantations (N18O 92′37N99O 32′78E), referring to coffee plantations without farmer management such as chemical fertilizer, pesticides, herbicides, and soil preparation. The average age of coffee plants was 17 years. Land use history was received by the interview from farmers.

At each location, five coffee plants of comparable diameter, stem, height, and canopy were selected. The average values of stem diameter and plant height were 76.6 mm and 2.64 m, respectively. The average diameter of the coffee plant canopy was approximately 200 cm. Soil samples were directly collected around the rhizosphere of coffee plants using a shovel, and then soil adhering to the root after shacking was manually collected. At each soil sampling, three points were sampled and pooled as a representative sample. In every soil sampled, root samples were directly taken at 1–15 cm of depth from three sections of fine roots and combined. After sampling, soil and plant samples were immediately placed in an ice box before being transported to the laboratory for further soil microbial analyses.

Root samples were washed three times in sterile and distilled water, respectively. Subsequently, their surfaces were sterilized with 70% ethanol for 5 min and then in 2.6% sodium hypochlorite solution for 5 min. Finally, the samples were washed ten times with sterile distilled water to remove the NaOCl. The coffee roots were cut into one mm lengths

and placed in a microcentrifuge tube contained 30 μL of sterile water. Root samples were softly ground by a stirring rod. Root suspension on the stirring rod tip was taken for streaking on tryptic soy agar (TSA) and incubated at room temperature for 7 days for bacterial colony formation. Colonies were selected and purified by subculturing three times on TSA media contained (g/L$^{-1}$) 3 g glucose, 2.5 g K$_2$HPO$_4$, 5 g NaCl, 3 g soy, 7 g trypton and 15 g agar with pH of 7.0 before their storage in a 40% glycerol solution at −80 °C until further analysis.

To isolate bacteria from rhizosphere soil, 10 g of soil was diluted in sterile water to concentrations of 10$^{-1}$, 10$^{-2}$, 10$^{-3}$, 10$^{-4}$, and 10$^{-5}$, and 100 μL of the sample was pipetted onto TSA medium, using the sterile water as a blank control, and incubated at room temperature for 7 days with other plates under the same conditions. The streak plate method was used to purify the bacteria after incubation. Pure cultures were stored on slants at 4 °C and in a 25% glycerol stock at −80 °C for future experiments.

### Screening and identification of bacterial isolates

All the bacterial isolates were qualitatively screened for nitrogen fixation on Burk's N free medium (pH 7.2) contained (g L$^{-1}$): 10 g glucose, 0.52 g K$_2$HPO$_4$, 0.41 g KH$_2$PO$_4$, 0.05 g Na$_2$SO$_4$, 0.2 g CaCl$_2$, 0.1 g MgSO$_4$ 7H$_2$O, 0.0025 g Na$_2$MoO$_4$ ·2H$_2$O, 0.005 g FeSO$_4$7H$_2$O and 15 g agar, and incubated at 30 °C for 7 days (*Weaver & Danso, 1994*).

All nitrogen fixing bacterial isolates were qualitatively tested for phosphorus solubilization and Indole 3-acetic acid (IAA) production. To determine the phosphorus solubilizing ability, the isolate was inoculated into 25 mL of pikovskaya broth (PVKB) contained: 5 g Ca$_3$(PO$_4$)$_2$, for 72 h at 25 °C with constant shaking at 125 rpm. After 72 h, Pikovskaya (PVK) broth was centrifuged at 4,500 rpm for 10 mins, and supernatants were taken and checked for soluble phosphorus content in the culture broth using the molybdenum blue method (*Murphy & Riley, 1962*). As a control, autoclaved PVK broth without bacteria was incubated under the same conditions as the other samples.

IAA production was determined by a colorimetric method (*Gordon & Weber, 1951*). The bacterial isolates were cultured in NA liquid medium (with L-tryptophan) in the dark for 3 days at constant shaking at 125 rpm. After 3 days, the culture medium was centrifuged at 4,500 rpm for 10 min. One mL of supernatants (clear zone) were mixed with two mL of Salkovskii reagent to develop a color, and the IAA production was measured using the absorbance at 530 nm with a UV spectrophotometer (*Ahmad, Ahmad & Khan, 2005*; *Bose, Shah & Keharia, 2013*).

### Small subunit ribosomal RNA (SrRNA) gene sequencing

S2-4a1 and R2-3b1 isolated from the coffee rhizosphere soil and root tissue, were chosen due to their ability to solubilize mineral phosphates and potassium. The genomic DNA of both bacterial isolates was extracted as described by *Murray & Thompson (1980)* and identified by amplifying and sequencing the 16S rRNA gene with the universal primers forward 785F (GGATTAGATACCCTGGTA) and reverse 907R (CCGTCAATTCMTTTRAGTTT). The nucleotide sequences were compared with those of the National Center for Biotechnology Information (NCBI) (http://www.ncbi.nlm.nih.gov) using the Basic Local Alignment Search Tool (BLAST) software.

## Shelf-life determination of the inoculant
## Carrier material and inoculum preparation

The carrier materials used in this study were perlite, vermiculite, diatomite and coconut coir dust. Perlite ($\sim$2-4 mm, $SiO_2 \geq 75$ wt%), vermiculite ($\sim$1–3 mm, $SiO_2 \geq 41$ wt%) and diatomite ($\sim$1−1.5 mm, $SiO_2 \geq 90$ wt%) were purchased from Shijiazhuang Huabang Mineral Products Co., Ltd. (China). While coconut coir dust ($\sim$0.25 mm) was provided by the factory in Thailand. To prepare the carrier materials, 50 g of each carrier material was placed into a plastic bag, which was then sterilized by autoclave at 121 °C for 1 h. The carrier material was sterilized again 3 days after the first sterilization.

In this study, the S2-4a1 and R2-3b1 isolates were chosen for the development of inoculants due to their higher ability to solubilize mineral phosphates and potassium and produce IAA as compared to other isolates. These selected isolates were incubated in molasses low yeast (MLY) at 12 rpm at 28 °C for 7 days until reaching a concentration of 1 $\times 10^8$ CFU mL$^{-1}$. After incubation, 15 mL of bacterial supernatants were put into the sterile carrier materials and incubated at 25 °C for 90 days. To measure the survival of bacteria, each inoculated carrier was sampled at 0, 7, 15, 30, 45 and 60 days after incubation. The pH, EC and moisture content of the inoculated carrier were also determined at sampling.

## Physico-chemical characteristics of carrier materials

Soil pH of all carriers were determined was measured by a pH meter using soil-water suspensions (1:5 v/v) and electrical conductivity (EC) by digital probe (Seven Excellence pH meter; Mettler Toledo, OH, USA).

## Coffee seedling growth promotion by the S2-4a1 and R2-3b1 isolates

Local Arabica coffee (Chiang Mai 80) seeds used in this study were surface-sterilized with 3% hydrogen peroxide for 3 min and subsequently rinsed with autoclaved distilled water five times. The sterilized seeds were sown into sterilized sand for 60 days after sowing (DAS), and then coffee seedlings of equal height were transplanted into black polyethylene bags (one plant per pot) filled with 1 kg of autoclaved media containing a mixture of soil and coconut coir dust based bioformulations (3:1 (w/w)) for the tests. The chemical analysis of the media for seedlings is presented in Table 1. The experiment was arranged in a completely randomized design with three replications, involving four treatments inoculated with the liquid formulation (S2-4a1 and R2-3b1 isolates) and an uninoculated control: (i) control (no inoculation), (ii) soil inoculated with coconut coir dust based bioformulations of S2-4a1 and (iii) R2-3b1 isolate and (iv) a mixture of S2-4a1 and R2-3b1 isolates. The inoculum preparation for S2-4a1 and R2-3b1 isolates was described above. The media treated with uninoculated MLY broth were used as controls. According to the mixture treatment, these bacteria were grown separately and then mixed in equal proportions at the time of application. All the experiments were performed in a greenhouse with 75–80% humidity and temperatures between 25 and 30 °C. Irrigation was supplied manually to keep the media moisture at the field capacity.
**Table 1** Phosphate and potassium solubilizing activities and IAA production of bacterial isolates isolated from rhizosphere soil (S2-4a1) and root tissue (R2-3b1).

| Isolate | Phosphorus solubilizing ability mg mL$^{-1}$ | Potassium solubilizing ability mg mL$^{-1}$ | IAA production mg mL$^{-1}$ |
|---------|------------------|------------------|------------------|
| S2-4a1 | $406.1 \pm 4.02$ | $18.86 \pm 1.08$ | $15.33 \pm 0.78$ |
| R2-3b1 | $301.7 \pm 4.02$ | $18.10 \pm 1.71$ | $16.54 \pm 1.02$ |

## Determination of plant nutrient uptake and media physico-chemical properties

The coffee plants were destructively harvested after 90 days of treatment to determine the dry mass of leaves, branches and roots, which were then dried in an oven with forced-air circulation at 55 °C until a constant mass was obtained. From each plant tissue, 0.5 g of dry biomass was subjected to nitric-perchloric acid digestion to analyze total nitrogen (N), phosphorus (P), potassium (K), calcium(Ca) and magnesium (Mg) content (*Bremner, 1996*). Total nutrient uptake was based on the nutrient content in the whole plants coffee biomass (roots, shoots, leaves) (Eq. (1)).

$$\text{Total nutrient uptake} = (\text{biomass} \times \text{nutrient concentration}(\%))/100. \tag{1}$$

## Determination of nutrient transfer factor

In this study, the nutrient transfer factor (NTF) was used as an indicator of the ability of the inoculated bacteria to improve nutrient uptake efficiency by transferring available nutrients into soil that can be directly absorbed by plants. The NTF was calculated by dividing the concentration of nutrients in the whole plants by the nutrient concentration in the media after the harvested plants. To determine the physico-chemical properties of the experimental media after harvest, the available phosphorus (P) of the media were determined by UV–VIS Spectrophotomete and the available K, Ca and Mg detection by ICP-OES (ICAP 6000 SERIES, Thermo Fisher Scientific, Waltham, MA, USA).

## Statistical analysis

Statistical analysis was conducted using analysis of variance (ANOVA). The least significant difference (LSD) was performed to compare the mean differences among treatments, and $P$ value $< 0.05$ was considered for the significant level. Linear regressions were carried out to determine the spatial variations in the pH and EC and their changes with times. The interaction effects of carrier type and time on the bacterial population, pH, and EC were quantified using a two-way ANOVA. The statistical software used was SPSS 19.0 (SPSS, Inc., Chicago, IL, USA).

# RESULTS

## Isolation and characterization of mineral solubilizing microbes

The S2-4a1 and R2-3b1 isolated from rhizosphere soil and root tissue were chosen due to their high abilities to solubilize phosphate and produce IAA as compared to other isolates.

The S2-4a1 and R2-3b1 showed their ability to solubilize phosphate with 407.1 and 301.7 mg L$^{-1}$, respectively (Table 1). However, negligible differences were observed for potassium solubilizing ability and IAA production between S2-4a1 and R2-3b1 isolates. Two isolates (S2-4a1 and R2-3b1) were identified using 16S rRNA gene sequencing for comparison with NCBI sequences using the BLASTn algorithm. The result found that S2-4a1 and R2-3b1 had a 99% similarity with *Bacillus megaterium* and *Bacillus* sp., respectively.

## The survival of bacteria

A suitable carrier material should be able to stimulate and provide an environment for microbial development and the survival of the inoculated bacteria over long-term storage. In this study, the different carriers (coconut coir dust, diatomite, vermiculite, perlite) inoculated with both bacterial S2-4a1 and R2-3b1 isolates were evaluated. The population of S2-4a1 inoculated in coconut coir dust, vermiculite and perlite and R2-3b1 inoculated in coconut coir dust, diatomite and vermiculite continuously increased during the first 30 days after inoculation (DAI) (Fig. 1). The highest population of S2-4a1 was observed in diatomite carrier, but there was no significant difference as compared to coconut coir dust and vermiculite ($P < 0.05$). While the highest population of R2-3b1 was found in the coconut coir carrier. At 45 and 60 DAI, there was no significant difference in the population of both S2-4a1 and R2-3b1 among carriers ($P < 0.05$), indicating that coconut coir dust could be used as an alternative carrier for developing the formulation of biofertilizer.

## Effect of carriers on pH and EC

The different carriers used in this study significantly influenced the pH and EC after bacterial inoculation ($P < 0.01$) (Table 2). The pH of diatomite, vermiculite and perlite carriers inoculated with both S2-4a1 and R2-3b1 was significantly higher than coconut coir dust during the incubation periods (Fig. 2). The pH ranges of coconut coir dust inoculated with S2-4a1 and R2-3b1 isolate were 6.25 to 6.99 and 6.29 to 6.90, respectively. While the highest electrical conductivity (EC) was observed in coconut carriers inoculated with both bacterial isolates as compared to other carriers (Fig. 3). The ranges of EC in coconut coir dust, diatomite, vermiculite and perlite inoculated with S2-4a1 isolate were 4.18 to 4.67, 0.72 to 0.99, 1.52 to 1.70 and 1.60 to 1.69 mS cm$^{-1}$, respectively. For the carrier inoculated with R2-3b1, the EC ranges were 4.14 to 4.68, 0.84 to 0.94, 1.36 to 1.60 and 1.70 to 1.78 dS cm$^{-1}$ for coconut coir dust, diatomite, vermiculite and perlite, respectively. During inoculation, only the coconut coir dust carrier inoculated with both bacterial isolates exhibited significant decreases in pH and EC ($P < 0.001$) in the present study (Figs. 2 and 3).

## Interaction between carriers and inoculation time

The two-way ANOVA indicated that carrier and time revealed significant effects on both S2-4a1 and R2-3b1 populations, pH and EC ($P < 0.05$), as shown in Table 3. Based on the S2-4a1 isolate, the interaction of carrier and time had significant effects on the S2-4a1 population ($P < 0.01$), pH ($P < 0.05$) and EC ($P < 0.01$). While the interaction of carrier and time had no significant effects on R2-3b1 population and pH. The impact of

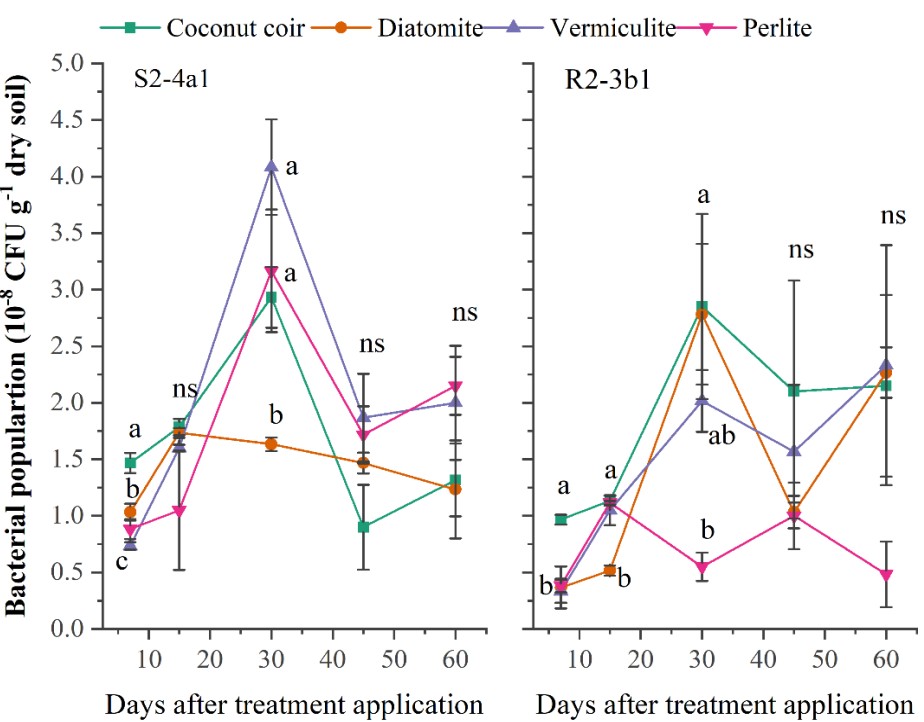

**Figure 1** **The population of two bacterial isolates (S2-4a1 and R2-3b1) inoculated in the different carrier materials over a storage period of 60 days.** The points sharing a common letter do not differ significantly at the $P < 0.05$. The different letters indica.

**Table 2** **Two-way analysis of variance of the bacterial population, pH and EC as affected by carrier and time after inoculation.**

| Variable | S2-4a1 | | | R2-3b1 | | |
|---|---|---|---|---|---|---|
| | Bacterial population | pH | EC | Bacterial population | pH | EC |
| F | | | | | | |
| Carriers | 3.800 | 338.89 | 667.69 | 4.900 | 160.97 | 3493 |
| Time | 22.760 | 2.600 | 0.520 | 6.880 | 3.540 | 5.690 |
| Carriers × Time | 3.510 | 3.630 | 2.560 | 1.390 | 1.180 | 5.240 |
| P | | | | | | |
| Carriers | 0.017[*] | 0.000[***] | 0.000[***] | 0.005[**] | 0.000[***] | 0.000[***] |
| Time | 0.000[***] | 0.051ns | 0.725ns | 0.000[***] | 0.015[**] | 0.001[**] |
| Carriers × Time | 0.001[**] | 0.001[**] | 0.013[*] | 0.211ns | 0.327ns | 0.000[***] |

**Notes.**
*, ** and *** show statistical significances at $P < 0.05$, 0.01, and 0,001, respectively.

environmental factors on both S2-4a1 and R2-3b1 populations varied with carrier types (Fig. 4). There were no any environmental factors (pH and EC) significantly correlated with both the S2-4a1 and R2-3b1 populations. However, the responses to carrier substrates of both S2-4a1 and R2-3b1 populations was significantly different. The S2-4a1 populations were significantly correlated with vermiculite and perlite, while the response of the R2-3b1 populations to carrier substrates was not observed in this study.

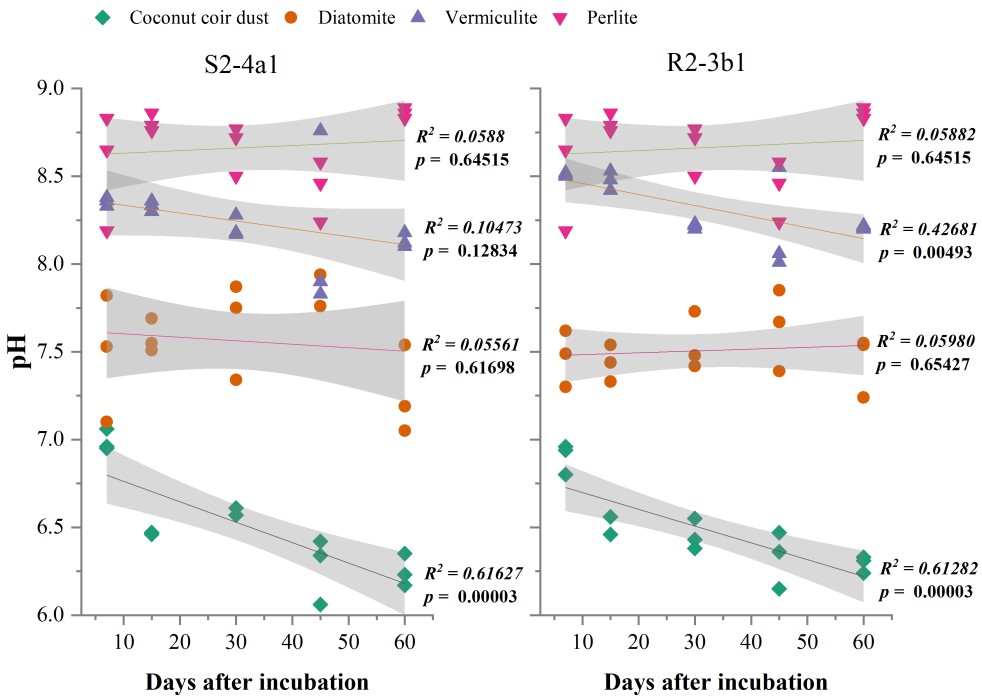

**Figure 2  Regression between the pH in the different carrier materials inoculated with S2-4a1 and R2-3b1 and days after inoculation.** The gray shading indicates the 95% confidence interval.

## Influence of isolated bacteria on biomass and nutrient uptake of coffee seedling under greenhouse conditions

In this study, two selected isolates (S2-4a1 and R2-3b1) were tested for their growth enhancement effect on coffee seedlings. The results showed that inoculated treatments increased the total biomass of the seedlings compared to the uninoculated control (Fig. 5). However, the isolate S2-4a1 increased biomass significantly more than the R2-3b1 isolate ($P < 0.05$). In addition, the inoculation of the S2-4a1 isolate increased the nutrient uptake of coffee seedlings (Fig. 6), which increased the N, P, K, Ca and Mg uptake by 48%, 48%, 40%, 80% and 56%, respectively, while the R2-3b1 isolate increased 21% (N), 18% (P), 18% (K), 22% (Ca) and 24% (Mg) when compared to the inoculated control. Interestingly, inoculation with the mixture of both tested isolates did not show a significant difference in biomass and nutrient uptake as compared to sole inoculation with the R2-3b1 isolate.

Pearson's correlation analysis was performed to determine the relationships between the plant P, K, Ca and Mg uptake, P, K, Ca, Mg transfer factor and plant biomass at 90 DAS (Fig. 7). The plant biomass was positively correlated with the plant P ($R^2 = 0.59$ $P < 0.01$), K ($R^2 = 0.95$ $P < 0.01$), Ca ($R^2 = 0.75$ $P < 0.01$) and Mg uptake ($R^2 = 0.89$ $P < 0.01$). The plant biomass attained a positive correlation with the K ($R^2 = 0.85$ $P < 0.01$), Ca (0.52 $P < 0.01$), and Mg (0.51, $P < 0.01$) transfer factor but was not significantly correlated with the P transfer factor ($R^2 = 0.22$, $P > 0.05$).

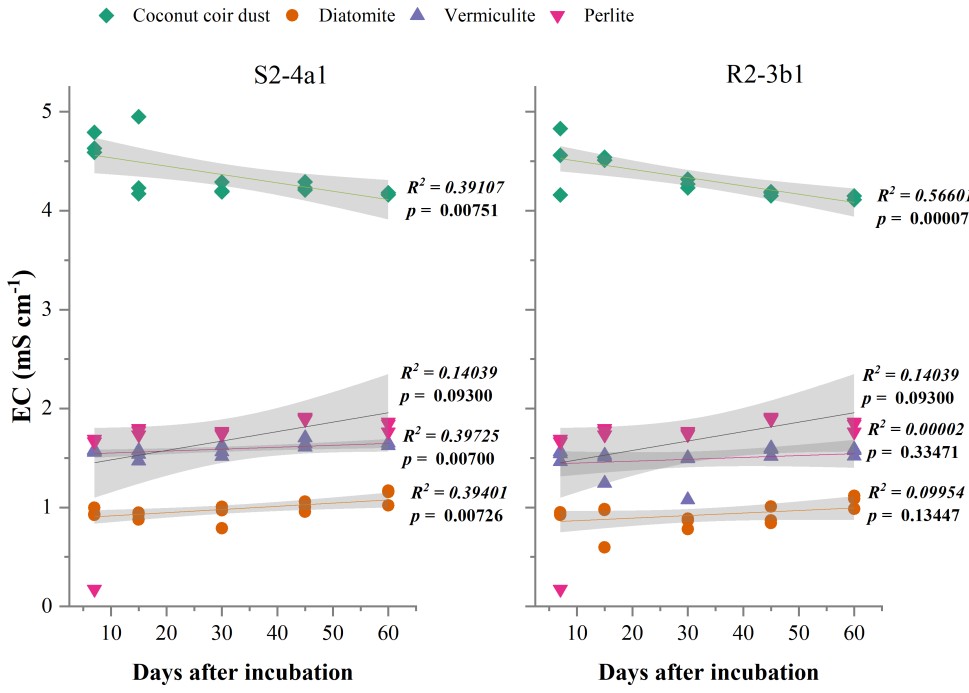

**Figure 3** Regression between the EC in the different carrier materials inoculated with S2-4a1 and R2-3b1 and days after inoculation. The gray shading indicates the 95% confidence interval.

**Table 3 Physico-chemical properties of the media used for production of coffee seedlings.**

| Physico-chemical properties | Value |
|---|---|
| pH (1:1) | 6.37 |
| EC (ds m$^{-1}$) | 1.58 |
| OM (%) | 3.83 |
| Available P (mg kg$^{-1}$) | 186 |
| Exchangeable K (mg kg$^{-1}$) | 429 |
| Exchangeable Ca (mg kg$^{-1}$) | 1,599 |
| Exchangeable Mg (mg kg$^{-1}$) | 421 |

## DISCUSSION

The primary step in developing a formulation of biofertilizer was to find a good carrier. Vermiculite and perlite have been widely used as good carriers for biofertilizers and biological control agents (*Albareda et al., 2008*; *Sangeetha, 2012*; *Sahu & Brahmaprakash, 2016*). The present study showed that coconut coir dust was equally effective as other substrates (diatomite, vermiculite, perlite) in maintaining high populations of both bacterial isolates (S2-4a1 and R2-3b1) after 90 days after inoculation. Similarly, *Corrêa, Sutton & Bettiol (2015)* discovered that the sustaining high densities of *P. chlororaphis* strains 63-28 and TX-1 were higher when formulated in coconut fiber than in the other formulation materials (talc, peat). In addition, cocopeat had the ability to support the

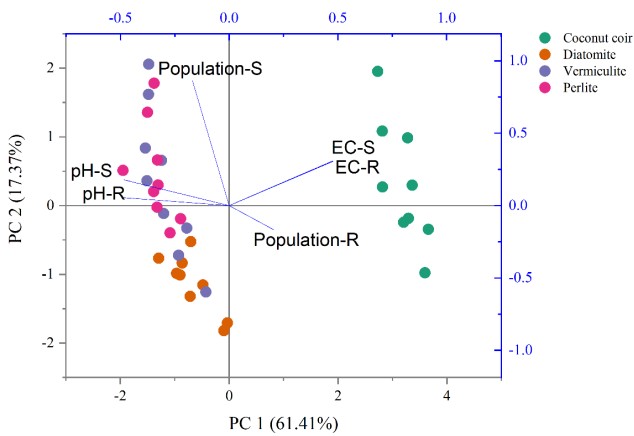

**Figure 4** Principal component analysis of pH, EC and population under the different carrier materials inoculated with S2-4a1 and R2-3b1.

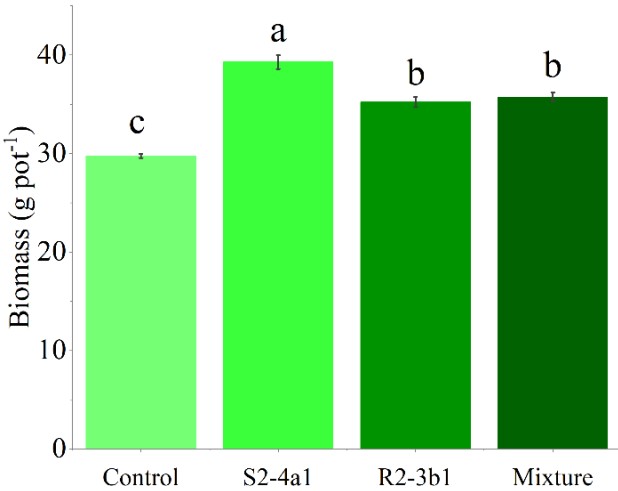

**Figure 5** Average coffee biomass inoculated to bacterial isolates. Same lowercase letter denotes no significant difference ($P < 0.05$). Bacterial mixtures (S2-4a1 and R2-3b1).

growth of two *Pseudomonas* sp. isolates (An-2-nali and Pn-2-kho) for up to 7 months with a population of $1 \times 10^4$ CFU g$^{-1}$ as compared to talc and farmyard manure carriers (*Verma et al., 2013*). Their results indicated that a smaller particle size of the cocopeat carrier led to an increased shelf-life (*Verma et al., 2013*). According to study of *Smith (1992)*, high nutrient content and water holding capacity (WHC) are considered the most important characteristics of good carriers. Thus, the high population of bacteria inoculated in coconut coir dust is probably attributable to its high nutrient content (EC), water holding capacity and small particle size in this study. The population of bacteria in all carriers inoculated with both bacterial isolates was satisfactory with inoculant quality standards, ranging from

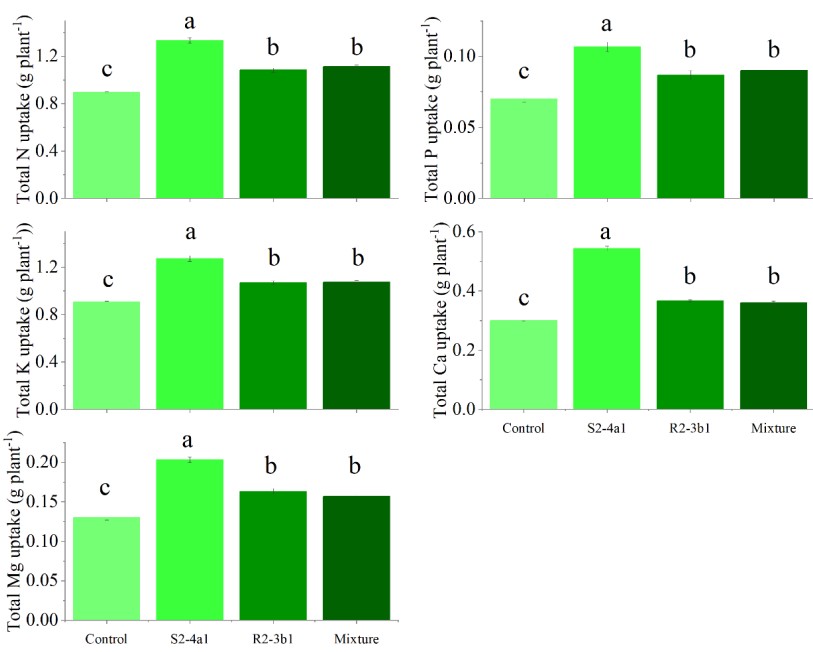

**Figure 6 Total nutrient uptake of coffee seeding inoculated with bacterial isolates.** Same lowercase letter denotes no significant difference ($P < 0.05$). Bacterial mixtures (S2-4a1 and R2-3b1).

$1 \times 10^7$ to $1 \times 10^9$ CFU g$^{-1}$ (*Malusa & Vassilev, 2014*). This finding proved that coconut coir dust is a suitable alternative carrier substrate for inoculant technology.

In the present study, the change in pH and EC of carriers inoculated with both bacterial isolates were significantly influenced by carrier substrates and inoculation times (Table 2). During the incubation time, the pH of diatomite, vermiculite and perlite carrier substrates inoculated with both bacterial isolates (S2-4a1 and R2-3b1) was above 7. While the pH of the coconut coir dust carrier ranged from 6.25 to 6.99, which is close to the optimum pH range for bacterial growth (*Nakkeeran, Fernando & Siddiqui, 2005*). Interestingly, the significant decrease in pH occurred only in coconut coir dust carriers inoculated with both bacterial isolates (Fig. 2). This decrease in pH might be due to the reducing of EC (the nutrient concentration) (Fig. 3) and water content available in accordance with the storage times, which was confirmed by the significant negative regression of both pH and EC with inoculation time in coconut coir dust carrier. Another reason is the increase in microbe metabolism activity in coconut coir dust carrier such as phosphate solubilization. These observations are similar to the findings of *Minaxi, Yadav & Saxena (2012)*, who observed that the decrease in pH of carriers is associated with the release of organic acids by phosphate solubilization during inoculant storage. These might be assumed that the microbe metabolism activity in coconut coir dust was higher than other carriers. However, we could not completely confirm that the decrease in PH and EC over time in this study was caused by inoculation or carrier effects due to the lack of each paired carrier without inoculation. The average EC of the coconut coir carrier inoculated with both bacterial

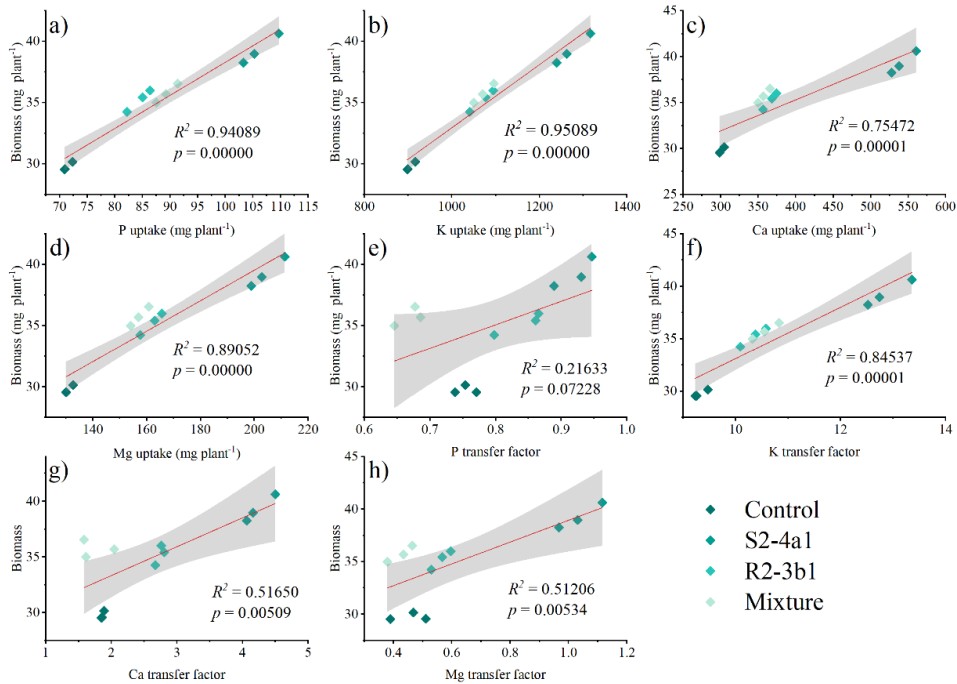

**Figure 7 Regression between the nutrient uptake (A, B, C, D) and the plant biomass.** Regression between the nutrient transfer factor (E, F, G, H) and the plant biomass after harvesting in the media inoculated with the different bacteria isolates (S2-4a1 and R2-3b1).

isolates ranged from 4.14 to 4.68 mS cm$^{-1}$, which was higher than the EC in the other carriers. Typically, the soluble salts of a carrier are represented by the EC concentration, and this concentration has a significant impact on the activities and survivability of inoculants (*Głodowska et al., 2016*). Although the pH and EC levels in the coconut coir dust carrier were nearly optimal for bacterial survival during the inoculation period, further optimization of storage conditions for long-term bacterial survival is required because the toxic substances may not have had enough time to act. The bacterial S2-4a1 isolate, which originated from rhizosphere soil of Arabica coffee, seems to cope greatly with various carriers, compared with the R2-3b1 isolate (Fig. 4). This might be due to its high adaptability. The findings of the current study suggest that the ecology in the areas where bacterial isolates were isolated may have affected the bacterial adaptability in carriers. Moreover, our results indicated that the responses of both bacterial isolates to the same carriers were significantly different. Therefore, these results are important for suitable carriers for each type of bacteria.

In addition, the present study revealed that inoculating bacterial isolates with coconut coir dust carriers significantly improved the growth of coffee seedlings (Fig. 5). This increase might be explained by the higher nutrient uptake promoted by two isolated bacteria (Fig. 6), as evidenced by the positive correlation between the biomass and nutrient uptake (Fig. 7). Similar findings in a study by *Cisneros et al. (2017)* revealed that *Kocuria* sp. and *Bacillus subtilis* were discovered in Colombia to be phosphate-solubilizing bacteria that improved the phosphorus availability in soil, resulting in the development of coffee seedlings.In

addition, the inoculation of *Pseudomonas*, *Bacillus*, *Azospirillum* and *Burkholderia* in vegetal, fruit and sugarcane plants, in which they showed beneficial effects in inoculated plants (*Oliveira, De Canuto & Urquiaga, 2006*; *Erturk, Ercisli & Cakmakci, 2012*; *Esitken et al., 2010*). Based on the obtained isolate-dependent differences, this study revealed a greater increase in the biomass and total nutrient uptake of coffee seedlings inoculated with the S2-4a1 isolate when compared to those inoculated with the R2-3b1 isolate (Figs. 5 and 6). Although the huge differences observed for phosphorus solubilizing ability between S2-4a1 and R2-3b1 isolates (Table 2), the lack of a significant difference in the correlation between the P transfer factor and plant biomass indicated that the increases in biomass in this study were not attributable to the P availability provided by inoculations (Fig. 7E). In contrast to the negligible differences for potassium solubilizing ability and IAA production between S2-4a1 and R2-3b1 isolates, the plant biomass was significantly positively correlated with other K transfer factors and other nutrients such as Ca and Mg (Figs. 7F, 7G and 7H), indicating that the S2-4a1 isolate might promote the growth of coffee seedlings by other mechanisms than its known capacity for phosphate and potassium solubilization. Because this experiment tested the isolated bacterial potential for improving coffee seeding based on biomass and nutrient uptake, it may be overlooked that these bacterial isolates stimulate growth *via* another mode of action. Similar to the findings of *Araújo et al. (2020)*, the rhizosphere fungus *Aspergillus niger* can function as a multifunctional microorganism for promoting the growth of coffee seedlings germinated in nurseries, in addition to its well-known nutrient solubilizing ability. In present study, we could not conclude that the ability for nutrient solubilization of S2-4a1 and R2-3b1 isolates was the only reason for increasing nutrient uptake and coffee biomass. Therefore, further research is required to identify phytohormones, such as indoleacetic acid and gibberellins, in relation to their mode of action in the aerial part and roots of plants, which provides better efficiency in nutrient uptake from soils. Interestingly, the present study found that the combination of isolates did not support the growth of coffee seedlings (Fig. 5). A similar finding was reported by *Giassi, Kiritani & Kupper (2016)*, who found that a mixture of bacterial isolates did not promote the growth of plants on citrus rootstocks as compared to an uninoculated control. Two factors may be related: first, the various modes of action of the included bacteria; second, the presence of competition among the microorganisms in the mixture (*Hibbing et al., 2010*). Our result seems to indicate that the mixed microorganisms used in our study were ineffective. Due to the high diversity of microorganisms in the coffee rhizosphere (*Caldwell et al., 2015*; *Muleta, Assefa & Granhall, 2007*; *Zhao et al., 2018*), further research is needed to figure out the optimal combination among diverse microorganisms to facilitate the selection of potential isolates that could increase the growth of coffee seedlings throughout the plant life cycle, leading to long-term benefits.

# CONCLUSIONS

The present study demonstrated that coconut coir dust can be used as a carrier for coffee bacteria in terms of nutrient abundance and ability to sustain growth and survival of inoculated isolates (S2-4a1 and R2-3b1) over time, which is a suitable alternative to diatomic, vermiculite, and perlite as carriers for bacterial inoculants. Consequently, employing coconut coir dust as PGPB carriers will enable the development of an efficient biofertilizer to meet Thailand's biofertilizer demand. Furthermore, bioformulation prepared from coconut coir dust inoculated with the S2-4a1 and R2-3b1 isolates can affect and promote the nutrient uptake and growth of coffee seedlings, indicating the potential of these microorganisms for application as coffee inoculants and thus leading to a sustainable agriculture concept.

## Funding

This research work was supported by Chiang Mai University. There was no additional external funding received for this study. The funders had no role in study design, data collection and analysis, decision to publish, or preparation of the manuscript.

## Grant Disclosures

The following grant information was disclosed by the authors:
Chiang Mai University.

## Competing Interests

The authors declare there are no competing interests.

## Author Contributions

- Yupa Chromkaew conceived and designed the experiments, performed the experiments, authored or reviewed drafts of the article, and approved the final draft.
- Thewin Kaeomuangmoon conceived and designed the experiments, authored or reviewed drafts of the article, and approved the final draft.
- Nipon Mawan analyzed the data, prepared figures and/or tables, authored or reviewed drafts of the article, and approved the final draft.
- Nilita Mukjang analyzed the data, prepared figures and/or tables, authored or reviewed drafts of the article, and approved the final draft.
- Nuttapon Khongdee analyzed the data, prepared figures and/or tables, authored or reviewed drafts of the article, and approved the final draft.

## DNA Deposition

The following information was supplied regarding the deposition of DNA sequences:
The 16SrRNA gene sequences are available at GenBank/European Molecular Biology Laboratory (EMBL)/DNA Data Bank of Japan (DDBJ): LC757023 and LC757024.

## Data Availability

The following information was supplied regarding data availability: Supplementary Files.

## Supplemental Information

Supplemental information for this article can be found online at http://dx.doi.org/10.7717/peerj.15530#supplemental-information.

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
