# Peer review of "Is coconut coir dust an efficient biofertilizer carrier for promoting coffee seedling growth and nutrient uptake?"

_PeerJ, doi:10.7717/peerj.15530_

## Round 0.1 · original submission · Minor Revisions

Authors, kindly find the comments from reviewers.

Please, provide the needed details, not only in the revised manuscript but also in the response to comments.

Also, kindly do well to strengthen the discussion of your results.

I look forward to the revised manuscript.

Thank you very much

Reviewer 1 ·

Basic reporting

no comment

Experimental design

no comment

Validity of the findings

no comment

Additional comments

I have reviewed the manuscript “Is coconut coir an eûective biofertilizer carrier for enhancing coffee seedling growth and nutrient uptake". The authors study how different carriers affect the survival of plant growth promotion bacteria and enhance the seedling growth growth and growth&nutrient uptake. The content is important for biofertilizer technology. The manuscript is well written, and the conclusion is supported by the experiment data.
I only have minor comments:
L1: suggest changing the title to a declarative sentence
L56: what is ASEAN
Table2: suggest adding p symbols after the values
Fig1: suggest changes to bar plots with raw points, the error bars in current figures overlap and hard to read.

Reviewer 2 ·

Basic reporting

1.1. Line 31: the sentence is repeated twice.

1.2. Line 50: is there a citation for the claims regarding the scale and distribution of coffee production in Thailand?

1.3. Line 85: in other literature sources, “CD” stands for “coconut coir dust”, not “coconut coir”. Does the carrier material in this study also come in the form of coir dust? Was the material supplied as dust, or supplied as fibers and processed into coir dust by the authors?

1.4. Line 96: previous studies are mentioned but not cited. Please provide examples of such studies, in which coconut coir was used to evaluate inoculate population after incorporation.

1.5. Line 124: probable typo - “L” instead of “μL”.

1.6. Line 161: no supplier or sourcing information is provided for any of the carriers.

1.7. Line 232: the identification of isolate R2-3b1 is reported only to the genus level (Bacillus), not down to species level. This is a perfectly reasonable result, since it is possible that this is a novel or poorly characterized species. However, if this is indeed the case, it should be explicitly pointed out.

1.8. Table 1: the authors report the performance of the two isolates for P and K solubilizing activity and IAA production. How do these numbers compare to other PGPB strains?

1.9. How would the lower soil pH of the inoculated coir affect plant growth? Is a lower pH a desirable or undesirable result for coffee plants?

1.10. Carrier particle size is referenced in the discussion as one of the most important factors in supporting bacterial populations. However, the particle size of the coir carrier used in this study was not reported. This makes the study more difficult to reproduce, while also limiting its applicability for anyone trying to implement this technique in coffee plantations.

Experimental design

2. The effect on plant growth and nutrient transfer were not studied across different carriers. The inoculated coconut coir carrier was only compared to an uninoculated control, not to inoculated perlite, vermiculite, diatomite, or any other inoculated carrier. This makes it difficult to evaluate the performance of coir - although it outperformed an inoculated control, it is not clear how it compares to other carriers.

Validity of the findings

No comments

Reviewer 3 ·

Basic reporting

Clear, unambiguous, professional English language used throughout.
• Language is mostly clear, understandable, and professional.
• The paper is well organized.
• Some English editing may be required to fix some awkward sentences.

Literature references, sufficient field background/context provided.
• Sufficient background/context is provided about the overall problem the research addresses: boosting sustainable coffee production in Thailand. Information is provided about the Thailand coffee agriculture industry, governmental policy, fertilization practices.
• Some background is provided about rhizospheric plant growth-promoting microorganisms, and the different functions they provide for host plants (i.e. mineral solubilization, N fixation, phytohormone production). However, I believe:
- At least a brief discussion of endophytic bacteria is warranted.
- A discussion of different existing methods of inoculum delivery would help this paper’s introduction. There is a mention of non-spore forming bacteria (L76-77), which could tie into this.
• Some background is provided about biofertilizer carriers. However, I believe:
- The scarcity or availability of different carrier materials should be further discussed, perhaps in the context of Thailand.

Professional article structure, figures, tables. Raw data shared.
• Overall article structure is good; figures and tables are presented clearly. Raw data is shared.

Self-contained with relevant results to hypotheses.
• The paper is self-contained with relevant results.

Experimental design

Original primary research within Aims and Scope of the journal.
• This is a research article within the Aims and Scope of PeerJ Life and Environment.
Research question well defined, relevant & meaningful. It is stated how research fills an identified knowledge gap.
• Research question/aims are well stated:
- Evaluating the effectiveness of coconut coir compared to other commercial carriers, for bacterial persistence.
- Evaluating the potential of bioformulations based on coconut coir for promotion of coffee seedling growth.

Rigorous investigation performed to a high technical & ethical standard.
• Data is replicated well in experiments, and appropriate statistical analyses.
• However, I would like the authors to elaborate on several of the following points, regarding their experimental design:
- For bacterial isolation from root tissue, were any steps taken to surface-sterilize roots after sampling, and before isolation?
- For isolate screening, the authors screened first for nitrogen fixation, and only isolates that passed this N-fixation screen proceeded to the P solubilization and IAA production screening. Is there a reason that the N fixation test was prioritized over the other two tests?

Methods described with sufficient detail & information to replicate.
• Mostly good, but please elaborate on the following:
- L109-110 Do the authors have information on the approximate diameter, stem, height, or canopy conditions of the coffee plants that were selected?
- L160 There appears to be no methods for this subheading, please check.

Validity of the findings

Impact and novelty not assessed. Meaningful replication encouraged where rationale & benefit to literature is clearly stated.
• Rationale for the study is clearly stated (improving sustainable coffee production via PGPM). This paper can benefit the literature surrounding the use of PGPM for coffee growth promotion, by proposing two new putative PGPM. Findings on the interactions between carrier material, inoculations, and pH and EC may also be important for the literature on biofertilizer delivery and persistence.

All underlying data have been provided; they are robust, statistically sound, & controlled.
• Data has been provided.
• Statistical analyses are adequately described and performed.

Conclusions are well stated, linked to original research question & limited to supporting results.
• Interpretations of findings are backed by citations of relevant literature. However, the authors should elaborate on:
- L342-345 More relevant to the discussion would be a discussion of previous literature that explores growth-promoting bacteria for, specifically, coffee plants. Are there any?
• Conclusions are reasonable and well stated.

Additional comments

• Figure 2 and Figure 3 legends: “95% confidential interval”, or “95% confidence interval”?
• L150 The term SrRNA has not been defined before this point, so it’s ambiguous. Consider rewording this subheading.
• L226-227 Please elaborate what you mean by “chosen due to their high abilities” or reword the sentence.
• Please be consistent with indenting the beginning of each paragraph.
• Please be consistent with capitalization (or not) of Arabica and Robusta.

---

## Round 0.2 · Minor Revisions

The authors have done great efforts, however, there are a couple of small issues that need their attention, as follows:

a) lines 37-38: " After inoculation with both bacterial isolates, pH and EC were significantly altered (P < 0.01) by different carriers". Do the authors mean "in different carriers"?

There is no measurement of the pH and EC of the carriers before inoculation, so the initial differences in pH and EC (figures 2 and 3) could be due to starting differences in these values in the different carriers. Some, but not all carriers change pH and EC over the course of the experiment.

b) There might be an experimental design flaw here: there is no mock inoculation of the carriers, so we do not know if the changes in pH and EC observed over the time course are due to the inoculation. This caveat needs to be mentioned in the discussion of these results (lines 325-352).

Authors, please kindly provide very detailed responses to these concerns, not only in the revised manuscript but in the reply to comments.
Thank you

Reviewer 1 ·

Basic reporting

no comment

Experimental design

no comment

Validity of the findings

no comment

Additional comments

The authors have addressed my comments

Reviewer 3 ·

Basic reporting

I thank the authors for taking the time to implement the suggested revisions. The language has been improved in requested areas, and the background literature has been expanded well. Specifically, the authors have included a brief discussion of endophytes and their roles in growth promotion.

Experimental design

Thank you for revising and elaborating the methods with regards to root surface-sterilization, rationale behind using N fixation ability as a primary screening, and sampling method for coffee plants. I believe the reproducibility of this paper has been improved.

Validity of the findings

Thank you for including a brief discussion of previous findings relating to coffee growth promotion via bacteria.

---

## Round 0.3 · accepted · Accept

I am satisfied with the authors' current revised manuscript. It is acceptable for publication. Thank you for considering PeerJ as your journal of choice and I am looking forward to your future scholarly contributions. Congratulations